# Molecular Biomarkers of Electroconvulsive Therapy Effects and Clinical Response: Understanding the Present to Shape the Future

**DOI:** 10.3390/brainsci11091120

**Published:** 2021-08-25

**Authors:** Elisabetta Maffioletti, Rosana Carvalho Silva, Marco Bortolomasi, Bernhard T. Baune, Massimo Gennarelli, Alessandra Minelli

**Affiliations:** 1Department of Molecular and Translational Medicine, University of Brescia, 25121 Brescia, Italy; elisabetta.maffioletti@unibs.it (E.M.); r.carvalhosilva@unibs.it (R.C.S.); massimo.gennarelli@unibs.it (M.G.); 2Psychiatric Hospital “Villa Santa Chiara”, 37142 Verona, Italy; marcobortolomasi.vr@gmail.com; 3Department of Psychiatry and Psychotherapy, University of Münster, 48149 Münster, Germany; Bernhard.Baune@ukmuenster.de; 4Department of Psychiatry, Melbourne Medical School, University of Melbourne, Parkville, VIC 3010, Australia; 5The Florey Institute of Neuroscience and Mental Health, The University of Melbourne, Parkville, VIC 3052, Australia; 6Genetics Unit, IRCCS Istituto Centro San Giovanni di Dio Fatebenefratelli, 25125 Brescia, Italy

**Keywords:** electroconvulsive therapy, treatment resistant depression, biomarkers, molecular mechanisms, neurotrophic system, inflammatory system, immune system, gene expression

## Abstract

Electroconvulsive therapy (ECT) represents an effective intervention for treatment-resistant depression (TRD). One priority of this research field is the clarification of ECT response mechanisms and the identification of biomarkers predicting its outcomes. We propose an overview of the molecular studies on ECT, concerning its course and outcome prediction, including also animal studies on electroconvulsive seizures (ECS), an experimental analogue of ECT. Most of these investigations underlie biological systems related to major depressive disorder (MDD), such as the neurotrophic and inflammatory/immune ones, indicating effects of ECT on these processes. Studies about neurotrophins, like the brain-derived neurotrophic factor (BDNF) and the vascular endothelial growth factor (VEGF), have shown evidence concerning ECT neurotrophic effects. The inflammatory/immune system has also been studied, suggesting an acute stress reaction following an ECT session. However, at the end of the treatment, ECT produces a reduction in inflammatory-associated biomarkers such as cortisol, TNF-alpha and interleukin 6. Other biological systems, including the monoaminergic and the endocrine, have been sparsely investigated. Despite some promising results, limitations exist. Most of the studies are concentrated on one or few markers and many studies are relatively old, with small sample sizes and methodological biases. Expression studies on gene transcripts and microRNAs are rare and genetic studies are sparse. To date, no conclusive evidence regarding ECT molecular markers has been reached; however, the future may be just around the corner.

## 1. Introduction

Major depressive disorder (MDD) is a highly prevalent psychiatric disorder worldwide, causing a great deal of disability, disease burden, and health costs [1,2]. Even though standard antidepressant treatments are often effective, unfortunately several patients with MDD do not respond sufficiently [2], and it is estimated that 18 to 55% of patients present treatment-resistant depression (TRD), which is defined as depression that does not remit or respond to two or more adequate trials of antidepressant therapy [3,4,5,6]. TRD is associated with illness chronicity, a reduced quality of life, and a higher risk for suicide. Especially for those patients, electroconvulsive therapy (ECT) is a highly recommended treatment strategy [7]. In general, response rates to ECT vary between 60 to 80% and the time to achieve clinical response is usually shorter in comparison to commonly used pharmacological treatments [7,8] making it one of the most potent and rapidly acting treatments for affective disorders. Moreover, ECT has been shown to reduce the hospital length of stay and the number of hospitalizations over a period of 3 years in patients receiving maintenance ECT sessions [9]. Even in patients not achieving remission shortly after ECT sessions, there is still a reasonable remission chance if treatment is continued and combined with psychotropic drugs [10]. In spite of the evidence showing that ECT is an effective intervention strategy and indication of its use in international guidelines, it is often underused, resulting in a high variability between different countries and regions with a general decline in utilization observable in American and European countries. This attitude is mainly due to stigma related to the perception of the treatment in its early years, when it was administered without the use of muscle relaxants and proper anesthesia, inadequate knowledge about its efficacy and security, patient choice, fear of cognitive impairments, and limited availability [11,12].

ECT seems to exert its antidepressant effect through its impact on regional brain structure and function via different neurotransmitter systems, inflammatory processes, and neurogenesis. However, despite the frequent and widespread use of ECT for more than 70 years, the molecular mechanisms underlying its efficacy remain largely unclear, as well as its precise mechanisms of action [7]. Moreover, understanding the therapeutic effects of ECT may also shed more light on the pathophysiological causes of severe MDD, and it can be the fundament for approaches aimed at individualizing therapy strategies.

## 2. Aim of the Review

Previous reviews on the molecular mechanisms underlying ECT effects have focused mainly on preclinical models [13,14], specific biological systems, such as immune and inflammatory cascades [15,16] or neuroendocrine responses [17], specific molecular levels, such as epigenetic mechanisms [18], or biological events related to possible cognitive impairments and other adverse effects [14].

On this basis, the goal of this narrative review is to provide a more comprehensive understanding of ECT biological mechanisms covering, as much as possible, the different biological systems and molecular levels involved in the ECT mechanism of action, comprising both preclinical models that support the study in humans and the translation into psychiatric practice, provided by existing clinical studies. Moreover, we have especially focused on potential biomarkers that can reliably predict ECT clinical outcomes, which can be remarkably useful for clinical practice.

## 3. Methods

In order to achieve the aims of the present narrative review, we focused on molecular studies conducted on ECT comprising its course and prediction of its outcomes, including both preclinical models on electroconvulsive seizures (ECS), an experimental analogue of ECT, and clinical studies.

Electronic searches were performed using MEDLINE/PubMed and Scopus databases combining the following keywords/search terms: “depression”, “major depressive disorder”, “MDD”, “electroconvulsive treatment”, “ECT”, “animal models”, “electroconvulsive seizures”, “ECS”, “molecular mechanisms”, “biological systems”, “neurotrophic system”, “inflammatory and immune systems”, “genetic”, “epigenetic”, “expression”, and “microRNAs”. Two of the authors (E.M., R.C.S.) independently reviewed the database to avoid mistakes in the selection of articles. The reference list of the studies, meta-analyses, and reviews on this issue were also reviewed in order to detect further publications. All studies, meta-analyses, and review articles on molecular biological markers of ECT in MDD published until March 2021 were included. Studies were selected if they met the following criteria: (a) being a molecular study conducted on ECT, regarding preclinical models or clinical studies in patients with MDD, (b) being in English, and (c) being an original paper published in a peer-reviewed journal. A total of 92 papers were selected from the literature for our narrative review, of which 43 were performed in preclinical models and 49 were done in clinical samples. We selected only preclinical studies presenting a direct link with or potential transferability to studies in humans.

After performing this search strategy and organizing the results of the most relevant studies, we structured this present review according to the main biological systems studied so far, from the most studied ones to the systems less investigated, including the ones with very little evidence and sparse results. Most of the molecular investigations conducted on ECT, concerning both its effects and the prediction of patients’ response, have been performed on candidate biomarkers and systems on the basis of the main pathogenetic hypotheses of MDD. The principal systems which have been studied providing significant results are represented by the neurotrophic and the inflammatory/immune ones. In addition to the neurotrophic and the inflammatory/immune ones, other biological systems have been more sparsely investigated, including the monoaminergic, glutamatergic, and the endocrine ones. Furthermore, we included studies concerning protein levels, gene expression changes, and genetic and epigenetic studies, when they were available, for the different biological systems. Genetic biomarkers of treatment response are of particular relevance since, compared to biomarkers constituted by proteins, metabolites, or gene expression levels, they could represent a more stable option, not being influenced by peripheral factors. The main findings, including genetic studies, are described below. 

## 4. Neurotrophic System

### 4.1. Protein and Gene Expression Studies

Regarding the main hypotheses and systems involved in MDD pathogenesis, the most studied system is represented by the neurotrophic one, formulated from experimental observations that revealed how antidepressant drugs are able to exert a modulatory action on neurotrophic factors or neurotrophins [19]. These molecules support the development and proper functioning of neurons and promote their resistance and regeneration in response to insults and stressful stimuli; antidepressants stimulate the activity of the genes encoding various neurotrophic factors [20,21]. 

At the beginning of the 2000s, first evidence indicated that ECS, an accepted experimental analogue of ECT employed in animal studies, enhances the neurogenesis in the hippocampus of rats, stimulating the development of new neurons [22], and the same thing was confirmed later in non-human primates [23]. Although the hippocampus is a highly stress-sensitive brain region, widely implicated in MDD pathogenesis, it is important to observe that the physiological function and pathological processes of hippocampus alone does not fully explain the complexity of symptoms of MDD and other neuropsychiatric conditions. The neurotrophic effects of ECS and ECT were suggested to be mediated, at least partially, by an increase of the intracerebral concentrations of the two neurotrophins brain-derived neurotrophic factor (BDNF) and vascular endothelial growth factor (VEGF) [24,25,26]. Subsequent investigations conducted in humans confirmed an enlargement of hippocampal volume in response to ECT [27,28]. The analysis of peripheral tissues, such as blood and serum/plasma, of patients with MDD showed an effect of ECT in increasing the levels of both BDNF and VEGF, suggesting that these molecular modifications could reflect brain changes [24] and further supporting that these neurotrophic processes are essential for the therapeutic effectiveness of ECT. 

BDNF represents the most abundant neurotrophin in the brain; during the development of the central nervous system, it promotes the growth of axons and the maturation of synaptic connections, whereas in the adult brain, it plays key roles in mechanisms of maintenance, stabilization, and neuroplasticity [29]. Moreover, there is evidence showing associations between BDNF and oxidative stress markers, with increased oxidative stress damage responses coupled with decreased BDNF levels, indicating that BDNF may play a protective role against oxidative damage in neurons [30]. This neurotrophin can also be found in the peripheral blood, where it is mostly stored in platelets [31]. In humans, BDNF levels are measured in the serum or plasma, since no imaging technique is available to accurately determine them in the brain. However, peripheral BDNF concentrations correlate with brain concentrations; therefore, they can reliably mirror brain conditions [32]. Reduced concentrations of BDNF have been detected in patients with MDD, and various antidepressant treatments have been described to restore physiological BDNF levels [33,34]. Studies in animals showed that ECS enhances the concentrations of BDNF in both the central nervous system and the periphery [24]. In particular, these findings showed that ECS induces significant increases of BDNF levels in the rat hippocampus [35,36,37,38] and that chronic ECS treatment also effectively increases hippocampal BDNF in the corticosterone animal model of depression [39]. In agreement with these findings, significant increases in gene expression were observed for BDNF and for other BDNF-related genes, such as *c-Fos*, *Egr1*, and *Neuritin 1*, after ECS stimulation in the hippocampus of rodents [40]. More recently, in animal models of depression induced by the chronic unpredicted mild stress procedure, ECS has been shown to reverse the low levels of BDNF in the hippocampus and effectively improve depressive-like behaviors [41], findings similar to the outcomes observed in depressed patients who received ECT treatment [42,43]. Regarding peripheral changes in BDNF in animals, some results showed a high positive correlation between central and serum BDNF [44]. Other authors studied the temporal expression profile of this neurotrophin in the hippocampus, prefrontal cortex, and serum, and found a positive correlation between central and peripheral levels [45]. In spite of these findings, the same correlations were not found in subsequent studies [46,47]. 

Coherently, studies on patients with MDD undergoing ECT have demonstrated an increase in peripheral BDNF levels, indicating this neurotrophin as an important mediator of the effects of this therapy [24,48,49]. In the context of the search for biomarkers that could help in the prediction of treatment effectiveness, which represents a particularly relevant issue for MDD and especially for TRD, BDNF has also been investigated for its potential to predict the outcome of pharmacological treatment when measured before the therapy or in its early phases. These studies indicated that early (1–2 weeks) increases in BDNF concentrations predict a better response to antidepressant drug treatment [50,51], whereas contrasting results came from studies that explored the possibility of exploiting baseline BDNF levels as predictors of drug treatment outcome [52,53]. Concerning the prediction of response to ECT, three studies are available and reported negative results concerning the predictive power of peripheral baseline BNDF [54,55,56]. 

VEGF is an angiogenic cytokine inducing vasopermeability, but it is also implicated in several brain processes regulating neuronal growth, differentiation, survival, and regeneration [57]. Changes in VEGF expression were found in rats’ hippocampus after acute and chronic ECS [26,58]. Also in rodents, ECS altered VEGF hippocampal and frontal cortex expressions, as well as its signaling components such as VEGFR2, required for cell proliferation, and its regulators, including mTORC1 [59]. In the last decade, several studies on peripheral tissues have been conducted to explore the involvement of VEGF in MDD, with important discrepancies across studies. Indeed, some indicated higher VEGF, some unaltered concentrations, and some lower levels in patients with MDD compared to non-affected subjects [60]. Inconsistent results also emerged from studies evaluating the effects of the most widespread treatments for MDD on circulating VEGF levels [61] and the possibility to exploit VEGF levels measured prior to treatment to predict its outcome [62,63]. ECT has been shown to enhance VEGF levels in TRD patients [64]; moreover, lower VEGF concentrations have been described to predict a lack of response to ECT, indicating that this neurotrophin could represent a predictive biomarker for treatment outcome [65,66]. A possible explanation for this could be related to the role played by VEGF in down-regulating the activity of the drug efflux transporter multiple drug resistance (MDR) P-glycoprotein at the brain–blood barrier. This leads to an increased brain content of MDR substrates, including antidepressant drugs. Since ECT has been reported to induce a transient breach of the brain–blood barrier and is usually administered to patients who are receiving antidepressant medication, elevated baseline VEGF levels could potentiate the action of ECT, augmenting the cerebral concentration of drugs [65]. 

Other neurotrophic factors which have been studied in relation to ECT include the glial cell-line-derived neurotrophic factor (GDNF) and the nerve growth factor (NGF). GDNF levels in the serum have been reported to increase following ECT in TRD patients; this effect was described as attributable to responders, since no significant change was observed in non-responders, suggesting that successful ECT is associated with enhanced GDNF [67]; in another study on rats subjected to ECS, decreased GDNF concentrations were noted in the hippocampus and the striatum, accompanied by increased levels of NGF in the frontal cortex [68]. 

### 4.2. Genetic Studies

Concerning the neurotrophic system, various single nucleotide polymorphisms (SNPs) in the *BDNF* gene, and in particular the functional polymorphism rs6265, also known as Val66Met since it causes a change of the encoded amino acid from valine to methionine, were studied in relation to ECT response, mainly with negative results [69]. A weak association was evidenced concerning rs11030101, for which patients with the TT genotype were more likely to benefit from ECT than those with the AT genotype [70]. In relation to VEGF, in one of the two studies in which lower baseline concentrations were observed to predict a lack of response to ECT, previously described, genetic variants influencing the levels of this neurotrophin have also been observed to predict treatment outcome. In particular, the A allele of rs78355601, located approximately 170 Kb downstream the *VEGF* gene and found as linked with lower VEGF concentrations, was associated with non-response to ECT [65]. Another study reported negative findings for another SNP, rs699947, found in the *VEGF* gene [71]. 

### 4.3. Epigenetic Studies

Most of the studies conducted to evaluate epigenetic modifications induced by electroconvulsive stimulation were done in animals, and specifically in rodents, suggesting that ECS exerts epigenetic effects both at the level of DNA methylation and histone acetylation. These involve mainly the neurotrophic system, through increased histone acetylation in the *BDNF* and related genes (*c-Fos* and *CREB*), and demethylation in the *BDNF* promoter [18]. A study assessing how posttranslational modifications of histones may be altered at specific gene promoters after acute or repeated ECS in rats’ hippocampus revealed positive correlations between H4 acetylation measures and mRNA levels of BDNF, c-fos, and CREB. Specifically, the authors observed a sustained down-regulation of c-fos transcription through H4 acetylation and a chronic upregulation of BDNF transcription via H3 acetylation, selectively at the BDNF P3 and P4 promoters [72]. Subsequent studies in humans examined epigenetic markers as predictors of treatment response for ECT, reporting that responders and remitters to ECT had significantly lower baseline serum methylation levels in the *BDNF* promoter [73]. Another study investigated the methylation in the promoter of p11, a molecule known to be involved in the up-regulation of BDNF and in serotonin signaling. In mice, responders to ECS had an increased pre-frontal cortex methylation in the p11 promoter, whereas TRD patients undergoing ECT showed similar findings in serum samples. Moreover, patients with higher baseline methylation had significantly lower depressive symptomatology after ECT treatment. In the same study, through the analysis of a separate patients’ cohort, p11 promoter methylation was observed to predict ECT response with medium levels of sensitivity (70%) and specificity (73%) [74]. 

In addition to epigenetic mechanisms, other important regulators of gene expression, also studied in relation to ECS and ECT responses, are represented by the microRNAs (miRNAs). These molecules represent a class of non-coding RNAs that regulate the translation of mRNAs and are known to be involved in key processes in central nervous system, including neuronal development, neurogenesis, and synaptic plasticity [75,76]. Alterations in these complex regulatory mechanisms have been linked to the development of MDD [77]. Changes in hippocampal miRNAs induced through early-life stress models of depression in rats, as well as outcomes of different antidepressant treatments, were investigated and ECS stimuli were able to reverse stress-induced changes in some miRNAs, including miR-598-5p, miR-217, miR-203, miR-211, miR-152, miR-1, and miR-204 [76]. ECS altered the levels of BDNF-associated miRNAs in rat brain and blood, specifically miR-212, known to be involved in the regulation of dendritic growth, dendritic plasticity, and spine density in the central nervous system, which makes this miRNA a potential biomarker for assessing the ECT response in depressed patients [78]. Studies in humans are lacking in this field, with the exception of one work demonstrating that the elevated peripheral blood baseline levels of miR-126-3p and miR-106a-5p, presented by seven patients with psychotic depression in comparison with health controls, were restored following ECT treatment [79]. In spite of the paucity of studies and reduced sample size, these preliminary findings demonstrate the potential involvement of miRNAs in the mechanisms of action of ECS stimuli and ECT. 

## 5. Inflammatory and Immune Systems

Another pathogenetic hypothesis of MDD based on which many studies on ECT have been designed is the inflammatory/immune one. In particular, the rationale for these investigations came from the observation of inflammatory and immune dysfunction in patients with MDD, such as increased levels of cytokines including the interleukins 6 (IL-6) and 1-Beta (IL-1β), the tumor necrosis factor-alpha (TNF-α), and the C-reactive protein (CRP), especially in TRD, for which ECT represents one of the most effective treatments. Moreover, higher baseline inflammation has been linked to a decreased response to pharmacological antidepressant therapies [80].

Some studies performed in animals pointed to several alterations in various aspects of the inflammatory and immune systems. It is known that glial cells can be subject to dynamic transformations in response to stress, reaching activated or reactive states, characterized by morphological alterations, proliferation, changes in gene expression, and secretion of inflammatory products [81,82]. Changes in glial cell morphology and enhanced expression of markers typical for reactive microglia, astrocytes, and NG2 expressing glial cells were observed in several rats’ brain regions following ECS sessions [81]. Other findings showed that antidepressant treatment with ECS can induce a strong proliferation of glial progenitor cells in rats’ hippocampus, as shown by increased NG2 expressing cell proliferation in hippocampal dentate gyrus following treatment [83]. Moreover, a decrease in microglial cell number in rats’ hippocampus was found after ECS administration, with a transient decrease observed 24 h after acute ECS and persistent decreases in animals receiving chronic ECS sessions, which lasted up to one month after the last ECS stimulus. This may indicate a possible differential response in acute and chronic ECS administration, valuable information for clinical ECT protocols, if further studies confirm this hypothesis [84]. However, in spite of these results, other findings did not demonstrate significant microglial changes following ECS administration [85]. In relation to neuroinflammatory cytokines, ECS was shown to increase the levels of proinflammatory cytokines like IL-1β and TNF-α in the hippocampus of depressed rats [86]. 

The application of ECT has been observed to induce an inflammatory-immune response in the short term, as part of an acute stress reaction, whereby repeated treatment rather induces a long-term down-regulation of the inflammatory and immune systems. Specifically, many studies conducted on patients’ peripheral tissues, mainly plasma, showed a rapid increase in the levels of the inflammatory mediators IL-1β and IL-6, accompanied by elevated concentrations of cortisol. However, at the end of the ECT treatment, a decrease in the concentration of both IL-6 and cortisol has been reported [15]. Moreover, the clinical improvement observed during ECT has been shown to be accompanied by a gradual and significant decline in TNF-α, eventually reaching levels comparable with those of healthy controls [87]. ECT has also been described to rapidly increase, after 15–30 min, the number and activity of leukocytes, including specific subpopulations such as granulocytes, monocytes, and natural killer cells [88,89]. In contrast, a decrease in leukocyte counts has been observed a few hours after the end of treatment [89,90]. ECT was also described to enhance the levels of neopterin, a protein produced by human monocytes/macrophages upon immune activation in depressed patients responding to the treatment but not in non-responders [91]. Other inflammatory mediators which have been studied in relation to ECT are the kynurenines. The kynurenine pathway plays a critical role in generating cellular energy and, since energy requirements are substantially increased during an immune response, this pathway represents a key regulator of the immune system; however, kynurenines exert behavioral effects that extend beyond those involved in inflammation and have been implicated in different psychiatric diseases including MDD [92]. A first study conducted on patients with both unipolar and bipolar disorders demonstrated an increase in the serum concentration of kynurenic acid in response to ECT, providing evidence that the mechanism of action of ECT could be mediated by the kynurenine pathway [93]. A subsequent investigation similarly showed that, prior to treatment with ECT, TRD patients had significantly lower levels of plasma kynurenic acid and quinolinic acid, another kynurenine metabolite, compared to non-affected controls, which increased significantly after ECT [94]. 

Concerning the prediction of treatment response, few studies are available, indicating an association between higher baseline levels of IL-6 and lower symptomatology at the end of treatment [95] and between moderately elevated baseline levels of the CRP and higher remission rates [96]. These results are in contrast with the evidence obtained for antidepressant drugs, indicating that inflammation contributes to poor treatment response, and their biological meaning is still unclear.

## 6. Monoaminergic System

### 6.1. Protein Studies

In relation to the monoaminergic hypothesis of MDD, postulating that the monoamines serotonin, noradrenaline, and dopamine are involved in the control of mood and that a decrease in their concentrations represents the basis for the development of the depressive pathology [97], over the past decades, investigations conducted in animal models have begun to depict the impact exerted by ECS on these neurotransmission systems, mainly the serotonergic and the dopaminergic ones [98]. Most of these early preclinical findings described an enhancement of the serotonergic neurotransmission, as reflected by the up-regulation of the postsynaptic receptors 5-HT1A and 5-HT2A in various areas of the brain [99,100,101,102], whereas subsequent human studies highlighted an effect of ECT in reducing the binding of serotonin to the same receptors in cortical and subcortical regions. However, there was no correlation between treatment outcome and the magnitude of changes in receptor binding [103,104].

Effects of ECS and ECT on the dopaminergic system have also been described, with consistency between data from animal and human studies; indeed, recent findings from studies conducted in humans have corroborated most of the earlier findings obtained in rodents and non-human primates, indicating that ECS/ECT induce an activation of the mesocorticolimbic dopamine system, involving different levels of regulation such as the release of dopamine [105,106] its binding to serotonergic receptors [107,108] and dopaminergic neurotransmission [109,110,111]. These molecular effects are supported by the observation of the improving clinical effects of ECT on typical dopaminergic functions, such as motivation, concentration, and attention [98]. 

Finally, the levels of homovanillic acid (HVA), a main monoamine metabolite, were found as increased in the cerebrospinal fluid and decreased in the plasma of depressed patients after ECT [105,106]; moreover, responders to the treatment presented higher baseline HVA levels compared to non-responders [106].

### 6.2. Genetic Studies

The main gene studied in relation to the monoaminergic system is the catechol-O-methyltransferase (*COMT*), encoding a methylation enzyme responsible for degrading catecholamines such as noradrenaline and dopamine. A common functional SNP in human *COMT* exon 4, a G to A substitution (rs4680), changes the encoded amino acid from valine to methionine. This missense mutation leads to a reduced activity of the COMT enzyme. This SNP has been suggested to represent a susceptibility variant for MDD [112,113] and to influence response to antidepressant treatment [114,115,116]. The results of a recent meta-analysis concerning the relation of rs4680 with response to ECT indicated that carriers of the G allele had a better response or remission to ECT. However, the authors pointed out that the number of studies about ECT is small and more research should be performed to confirm this result [117]. Significant associations with ECT outcome were also reported for polymorphisms in genes encoding dopaminergic receptors. In particular, concerning two SNPs in the dopamine D3 receptor (*DRD3*) gene, namely rs37322790 and rs3773679, the T and G allele, respectively, were observed to confer a better response/remission [118], whereas another study demonstrated that depressed patients heterozygous (CT) for rs6277, located in the dopamine D2 receptor (*DRD2*) gene, were more likely to remit compared to CC homozygotes [69]. Finally, the long/short polymorphism in the promoter of the serotonin transporter gene (5-HTTLPR) was implicated in response to ECT in combination with rs2242446, found in the noradrenaline transporter (NET) [119]. However, these investigations were conducted on quite small cohorts, including approximately one to a few hundred patients; findings should therefore be considered preliminary, and large-scale confirmatory studies are needed [120].

## 7. Glutamatergic System

### 7.1. Protein and Gene Expression Studies

In relation to glutamatergic system, ECS was shown to down-regulate the expression of GLT-1, a glial glutamate transporter, and to elevate the concentration of glutamate in the hippocampus of rats submitted to the chronic unpredictable mild stress model of depression [86]. Moreover, as previously reported, repeated ECS reduced the expression of genes related to N-methyl-D-aspartate (NMDA) receptor signaling, including the glutamate receptor, in rats’ frontal cortical regions [121]. The metabotropic glutamate receptor (GluR1) and NMDAR1 glutamate receptor subunit demonstrated significant changes in rat hippocampus after acute ECS, indicating that glutamate signaling pathways may mediate the effects of ECS [26]. ECS also produced significant elevated glutamate levels in rat hippocampus [122]. In animal models of depression, the glutamate content in the hippocampus was elevated and NMDA-NR2B, a type of NMDA receptor, was down-regulated, in comparison to controls. These effects were reversed after ECS stimuli, along with an amelioration observed in depressive symptoms [123]. Moreover, in rats submitted to olfactory bulbs removal to establish a depression model, the hippocampal glutamate levels and the hyperphosphorylation of tau protein, related to glutamate metabolism, remarkably increased after ECS. The changes in tau protein hyperphosphorylation were correlated with the electric current and duration of ECS stimuli [124].

With respect to clinical studies, some researchers investigated the effects of ECT on the glutamatergic system in depressed patients. A significant increase in N-acetylaspartate, an amino acid related to neuron functionality, and in glutamine/glutamate levels, assessed by proton stimulated echo acquisition method spectroscopy, was found in TRD patients who responded to ECT in comparison to non-responders [125] and similar findings were shown by other authors [126]. In this last study, a restoration of glutamine/glutamate levels similar to those presented by healthy controls was observed. Moreover, ECT normalized the reduction of glutamate levels in the anterior cingulate cortex in patients with MDD [127]. 

### 7.2. Genetic Studies

There is some evidence of genetic findings showing correlation with ECT response in the glutamatergic system. A study demonstrated that variants in the glutamate receptor ionotropic kainate 4 (*GRIK4*) gene, which is known to be associated with learning, memory and regulation of cognitive behaviors and mood [128], were associated with ECT response in a cohort of TRD patients and in a sample of depressed patients with bipolar disorder who were resistant to pharmacological treatments. More specifically, individuals with the G allele or with the GG genotype of the *GRIK4* polymorphism rs11218030 presented a worse response to ECT in comparison to subjects with the A allele and with the AA genotype, with the G allele carriers presenting five times the risk of non-response to ECT, compared to the AA homozygotes one month after the end of treatment. Moreover, subjects with the GG rs1954787 genotype and the rs4936554 A allele carriers showed a double risk of non-response to ECT. These findings point to putative *GRIK4* variants that can contribute to TRD and modulate the response to ECT treatment [129]. 

### 7.3. Epigenetic Studies

Epigenetic changes inducing gene expression regulations in the glutamatergic system were found in animal studies. Repeated ECS increased the expression of a histone deacetylase (HDAC2) in rats’ frontal cortical regions and reduced the expression of genes related to NMDA receptor signaling, including c-fos, glutamate receptor, and neuritin, with decreased histone acetylation levels of H3 and/or H4 and increased HDAC2 occupancy in the promoter regions of these genes after repeated treatment [121]. These findings suggest epigenetic events playing a role in the biological mechanisms involved in ECS treatment, with repeated seizures inducing gene expression alterations through histone modifications. 

## 8. Endocrine System

Among the systems proposed as involved in the antidepressant effects of ECT, attention has also been focused on the endocrine one. In addition to the increase of cortisol, previously mentioned in relation to the implication of the inflammatory/immune system, ECT has also been shown to exert enhancing short-term effects on the release of other hormones involved in the hypothalamic–pituitary–adrenal axis, including adrenocorticotropin (ACTH), prolactin, and vasopressin. These increases appear to be normalized within 1-h post-treatment [130,131,132,133], whereas studies in rodents indicate a longer-lasting increase in plasma ACTH, with elevated levels until 24 h post-treatment [134]. Moreover, as already described, ECT induces an increase in the blood–brain barrier permeability; thus, thanks to their elevated hematic levels, these hormones can be more easily absorbed and distributed in the central nervous system. Dynamic function tests assessing the integrity of the stimulatory and feedback regulation of the neuroendocrine system have also provided evidence of a restored neuroendocrine activity following ECT. In particular, the dexamethasone challenge test has been employed to investigate changes in the hypothalamus–pituitary–adrenal (HPA) repression status after ECT, leading to the observation of a reduced cortisol response to dexamethasone after the treatment compared to before it [135,136]. This has been hypothesized to mediate the remission of depressed mood and the restoring of vegetative functions, such as sleep, appetite, and sexuality, induced by ECT [17].

## 9. Oxidative Stress System and Mitochondrial Bioenergetics

Regarding oxidative stress metabolism and mitochondrial bioenergetics, many studies are available to date. Although being part of the normal metabolism, reactive oxygen species (ROS), when produced in excess as a result of chronic or traumatic stress, have the potential to cause tissue injuries through various mechanisms including lipid peroxidation, DNA damage, and enzyme inactivation [137]. Chronically increased ROS production levels is one of the mechanisms involved in the pathophysiology of MDD, adding to the body of evidence pointing to stress-associated biological damages in the body [7]. Evidence indicates that MDD is associated with an impaired bioenergetic supply and several changes in mitochondrial metabolism [7].

In order to investigate the consequences related to oxidative damages and antioxidant enzyme activities in the course of ECS, some animal studies were performed. Following single or multiple ECS, a decrease in oxidative damage parameters was demonstrated, represented by thiobarbituric acid reactive species as a measure of lipid peroxidation and protein carbonyls, in the rat hippocampus, showing that there is an increase in antioxidant enzyme activities after ECS [137]. The same authors found a decrease in lipid peroxidation and protein carbonyls in the rat hippocampus, cerebellum, and striatum after a single or multiple ECS and, in contrast, an increase in lipid peroxidation in the cortex, implying that ECS may cause a reduction of oxidative damage in the hippocampus, striatum, and cerebellum, but an increased oxidative effect in the cortex [138]. Other authors found increases in rat hippocampal and cerebellar superoxide dismutase (SOD) and glutathione peroxidase (GPX) activities, inferring increased antioxidant activity, after single ECS-induced seizures [139]. Contrary to these findings, exposure to ECS induced a decrease in SOD and GPX activity, in various rats’ brain regions, including the frontal cortex and the hippocampus, that persisted 48 h after the stimulation [140]. Furthermore, ECS caused a structure-related occurrence of delayed oxidative damage specially after multiple ECS in the rat hippocampus and striatum [141]. Specifically, while lipid peroxidation products and protein carbonyl levels increased, showing protein damage, the activities of SOD and catalase (CAT) decreased in various brain regions, including the hippocampus and striatum, showing a general increase in oxidative stress and a decrease in antioxidant enzyme activities following ECS. A possible explanation for this is that the increase in oxidative stress markers could reflect a simultaneous increase in metabolic rates or activation of metabolic pathways relevant to the therapeutic effects of ECS [141]. More recently, a decrease in mitochondrial respiration and an increase in RNA oxidation was demonstrated, as measured by 8-oxo-7,8-dihydroguanosine, in the rat brain tissue after chronic ECS, revealing an increased oxidative stress environment following ECS [142]. Overall, in spite of contrasting findings, the above-mentioned literature points to important metabolic regulations in the oxidative stress system following ECS. Despite some clarifications provided by recent studies on the mechanisms of action of ECS, the causes of oxidative stress and further adaptive antioxidant systems remains to be discovered, with the future possibility of translating these findings to clinical applications in MDD.

Regarding clinical investigations, one study revealed that ECT produced a significant reduction in the serum total oxidant status values and a significant increase in total antioxidant status in patients with MDD, showing that ECT did not increase oxidative stress in these patients [143]. Other authors studied ECT effects on serum markers involved in oxidative regulations, such as malondialdehyde (MDA), nitric oxide (NO) levels, and xanthine oxidase and SOD activities, in patients with MDD and bipolar disorders, and showed that SOD activity decreased after ECT, inferring decreased oxidative responses after this treatment [144].

In sum, further studies providing evidence of ECT effects at the oxidative stress and mitochondrial bioenergetic level will improve our understanding of the biological mechanisms underlying the pathophysiology of MDD, leading to potential new therapeutic approaches that may specifically target these systems.

The main findings concerning ECS/ECT mechanisms of action in relation to the different biological systems described in the above sections are summarized in Table 1, Table 2, Table 3, Table 4, Table 5 and Table 6 below.

## 10. Conclusions and Future Directions

Decades of research performed to elucidate the mechanism underpinning ECT outlined a vast field of study and found multiple molecular mechanisms that involve numerous intricate biologic processes, including alterations in neuroplasticity, levels of various neurotrophic factors and neurotransmitters, immune and inflammation mechanisms, neuroendocrine function, and epigenetic processes.

In particular, ECT improves neuroplasticity and increases neurotrophic factors such as BDNF and VEGF, but no clear association with clinical response was found. Indeed, whereas for BDNF no correlation of basal concentrations with ECT outcome was demonstrated, for VEGF there is evidence of a possible role as a predictive biomarker of ECT response. Moreover, more consistent data presented highlight that an acute immuno-inflammatory response occurs immediately following an ECT session, and this appears to be reversed after the end of the treatment course. In relation to the previously mentioned neurotrophic effects of ECT, it has been also hypothesized that the long-term reduction in inflammation could promote an enhancement in the levels of neurotrophic factors, such as BDNF. In addition, some biological effects of ECT may involve transient alterations in neuronal activity, as well changes in transcriptional regulations, possibly through epigenetic mechanisms, leading to long-lasting effects on synaptic and structural plasticity. ECT can transcriptionally target epigenetic enzymatic machinery, promoting changes in chromatin remodeling and persistent effects on gene regulation. 

Despite the evidence reported, the exact molecular biological mechanisms of ECT are unclear and inconsistent and research findings preclude from drawing firm inferences. This is attributable to lacunae in present literature, such as a lack of homogeneity in research methodology and small to moderate sample sizes. Most studies investigated effect on proteins with the possibility of confounding factors, the few genetic studies performed so far had a candidate gene approach and no genomic study of ECT response is available to date; expression studies are rare, investigating both RNA or miRNA levels, as well as epigenetic studies. Moreover, small sample size often does not allow post hoc analysis to identify possible interaction between molecular biomarkers and clinical features as well as neurophysiological measures (i.e., seizure threshold and duration) that might have an influence on the clinical effectiveness of ECT. Consequently, the cause–effect relationship between findings and therapeutic effects of ECT could not be established with absolute certainty.

Since ECT is the most effective treatment for severe and refractory depression but, at the same time, one of the most controversial and misunderstood treatments, increasing the knowledge about its mechanisms could help reduce its stigmatization and false beliefs about it. For example, ECT has always been accused of being a coercive, unethical, and dangerous modality of treatment and the dangerousness of ECT has been mainly attributed to its claimed ability to cause brain damage. However, biological research indicates clearly that there is a lack of evidence at present to suggest that ECT causes brain damage and, on the contrary, studies on neurotrophic factors have definitively shown that ECT stimulates neurogenesis and increases neuroplasticity.

For all the reasons just described, future studies in these fields are needed to identify the mechanisms responsible for high response following ECT in order to optimize and personalize the treatment. Furthermore, additional studies in this area may lead to better understanding of treatment-resistant mood disorders’ pathophysiology, and may help in the development of new and improved pharmacological treatments. In this regard, although epigenetic findings are still exploratory, epigenetics could represent a novel kind of biomarker for ECT treatment response, also in combination with more established predictors; certainly, more research is needed before reliable conclusions can be reached, and other biological systems implicated in the pathophysiology of MDD, aside from BDNF signaling, represent possible targets for future epigenetic studies.

Moreover, aside from the hypothesis-driven approaches in the different biological systems outlined in this review, broader untargeted analyses derived from genome-wide associated studies and from metabolomics, transcriptomics, or proteomics studies in ECT, which are mainly currently uncovered, would be a very promising field of research in future studies. Furthermore, additional insights in the molecular mechanisms of ECT and potential biomarkers acquired through these broader multi-omics approaches could be translated into other correlated research domains, such as other neuropsychiatric disorders, including epilepsy and schizophrenia. Adding to the development of advanced imaging technology, including structural and functional magnetic resonance imaging, with advanced perfusion and spectroscopic techniques, the resulting combined molecular, structural, and functional information may add promising perspectives of integrated research in the future. 

To overcome the above-mentioned limitations, future research focusing on molecular mechanisms of ECT need to achieve homogeneous methodology in a larger study population. To this aim, in late 2017 the first large-scale international consortium of ECT genomics, namely the Genetics of Electroconvulsive Therapy International Consortium (Gen-ECT-ic) [145] was formed. Gen-ECT-ic intends to organize the largest clinical and genetic collection to study the genomics of severe depression from the affective disorder spectrum (unipolar depression, bipolar disorder) and response to ECT, aiming for 30,000 patients worldwide using a genome-wide association study approach. It is hoped that with the effort of all these international researchers in the near future, it could be possible to obtain interesting data concerning the genomic underpinnings of severe depression and ECT outcome with relevant translational effects.

## Figures and Tables

**Table 1 brainsci-11-01120-t001:** Main findings of ECS/ECT biological mechanisms of action related to the neurotrophic system.

Kind of Study	Protein/Gene Studied	Population Studied	Main Findings	References
**Protein and gene expression studies**	BDNF	Animal models, including depression models induced by corticosterone administration	ECS increased hippocampal BDNF levels.	[35,36,37,38,39]
	BDNF	Animal models (depression model induced by the chronic unpredicted mild stress procedure)	ECS reversed low levels of BDNF in the hippocampus of depressed rats and improved depressive-like behaviors.	[41]
	BDNF and related genes	Animal models	ECS increased gene expressions of BDNF and BDNF related genes (c-Fos, Egr1, Neuritin 1).	[40]
	BDNF	Clinical samples	ECT increased peripheral BDNF levels.	[48,49]
	BDNF	Clinical samples	Peripheral BDNF levels did not predict ECT responses.	[54,55,56]
	VEGF	Animal models	Acute and chronic ECS sessions induced changes in VEGF expression in rats’ hippocampus.	[26,58]
	VEGF and signaling components	Animal models	ECS altered VEGF hippocampal and frontal cortex expressions.	[59]
	VEGF	Clinical samples	ECT enhanced peripheral VEGF levels in patients with TRD.	[64]
	VEGF	Clinical samples	Lower peripheral VEGF concentrations predicted a lack of response to ECT.	[65,66]
	GDNF and NGF	Animal models	ECS decreased GDNF concentrations in the hippocampus and the striatum and increased NGF levels in the frontal cortex.	[68]
	GDNF	Clinical samples	ECT increased peripheral GDNF levels in patients with TRD who responded to the treatment.	[67]
**Genetic studies**	*BDNF* gene	Clinical samples	Patients with the TT genotype for the SNP rs11030101 were more likely to benefit from ECT than those with the AT genotype.	[70]
	*VEGF* gene	Clinical samples	The A allele of rs78355601 was linked with lower VEGF concentrations and was associated with non-response to ECT.	[65]
	*VEGF* gene	Clinical samples	The VEGF 2578 C/A polymorphism was associated with TRD, but was not associated with treatment response to ECT.	[71]
**Epigenetic studies**	*BDNF* and related genes	Animal models	Positive correlations between H4 acetylation and mRNA levels of BDNF, c-fos, and CREB were found after ECS.	[72]
	*BDNF* gene	Clinical samples	ECT responders and remitters presented lower baseline methylation levels in the *BDNF* promoter.	[73]
**MicroRNA studies**	miR-212	Animal models	ECS altered the levels of BDNF-associated miRNAs in rat brain and blood, specifically the miR-212.	[78]
	miR126-3p and miR-106a-5p	Clinical samples	Patients with psychotic depression with elevated baseline levels of miR126-3p and miR-106a-5p, in comparison to controls, showed a restoration after ECT.	[79]

BDNF: Brain-derived neurothrophic factor; CREB: cAMP response element-binding protein; ECS: electroconvulsive seizure; ECT: electroconvulsive therapy; GDNF: glial cell-line-derived neurotrophic factor; miR: microRNA; NGF: nerve growth factor; SNP: single nucleotide polymorphism; TRD: treatment-resistant depression; VEGF: vascular endothelial growth factor.

**Table 2 brainsci-11-01120-t002:** Main findings of ECS/ECT biological mechanisms of action related to the inflammatory and immune systems.

Kind of Study	Protein/Gene Studied	Population Studied	Main Findings	References
**Protein studies**	Inflammatory markers	Animal models	ECS induced changes in glial cell morphology and enhanced expression of markers of reactive microglia, astrocytes, and NG2 glial cells.	[81]
	Inflammatory markers	Animal models	ECS induced proliferation of glial progenitor cells in hippocampus.	[83]
	Inflammatory markers	Animal models	ECS induced decreases in microglial cell numbers.	[84]
	Inflammatory markers	Animal models (depression models)	ECS increased the levels of IL-1β and TNF-α in the hippocampus of depressed rats.	[86]
	Inflammatory markers	Clinical samples	ECT increased the levels of IL-1β, IL-6, and cortisol.	[15]
	Inflammatory markers	Clinical samples	ECT increased the number and activity of leukocytes.	[88,89]
	Inflammatory markers	Clinical samples	ECT decreased leukocyte counts a few hours after ECT treatment.	[89,90]
	Inflammatory markers	Clinical samples	ECT enhanced neopterin, involved in immune activation, in patients with MDD who responded to ECT.	[91]
	Inflammatory markers	Clinical samples	ECT increased serum concentration of kynurenic acid, involved in immune responses.	[93]
	Inflammatory markers	Clinical samples	ECT increased plasma levels of kynurenic acid and quinolinic acid in patients with TRD.	[94]
	Inflammatory markers	Clinical samples	There was an association between higher IL-6 and lower symptomatology after ECT and between elevated baseline CRP and higher remission after ECT.	[95,96]

CRP: C-reactive protein; ECS: electroconvulsive seizure; ECT: electroconvulsive therapy; IL: interleukin; TNF: tumor necrosis factor; TRD: treatment-resistant depression.

**Table 3 brainsci-11-01120-t003:** Main findings of ECS/ECT biological mechanisms of action related to the monoaminergic system.

Kind of Study	Protein/Gene Studied	Population Studied	Main Findings	References
**Protein studies**	Serotonergic receptors	Animal models	ECS enhanced serotonergic neurotransmission, with up-regulation of 5-HT1A and 5-HT2A in various brain area.	[99,100,101,102]
	Serotonergic receptors	Clinical samples	ECT reduced the binding of serotonin to 5-HT1A and 5-HT2A in cortical and subcortical region.	[103,104]
	Monoamine metabolites	Clinical samples	Homovanillic acid, a monoamine metabolite, increased in cerebrospinal fluid and decreased in plasma in depressed patients after ECT treatment.	[105,106]
**Genetic studies**	*COMT* gene	Clinical samples	G allele carriers for the SNP rs4680 in the *COMT* gene had a better response or remission to ECT.	[117]
	*DRD3* gene	Clinical samples	SNPs in *DRD3* conferred a better response/remission after ECT.	[118]
	*DRD2* gene	Clinical samples	Patients with MDD and heterozygous (CT) for rs6277 in the *DRD2* gene were more likely to remit than CC homozygotes.	[69]
	Serotonin receptor and noradrenaline transporter	Clinical samples	5-HTTLPR was related to response to ECT in combination with rs2242446 in the noradrenaline transporter.	[119]

5-HT1A: serotonin 1A receptor; 5-HT2A: serotonin 2A receptor; 5-HTTLPR: serotonin-transporter-linked promoter region; *COMT*: catechol-O-methyltransferase; *DRD2*: dopamine D2 receptor; *DRD3*: dopamine D3 receptor; ECS: electroconvulsive seizure; ECT: electroconvulsive therapy; SNP: single nucleotide polymorphism; TRD: treatment-resistant depression.

**Table 4 brainsci-11-01120-t004:** Main findings of ECS/ECT biological mechanisms of action related to the glutamatergic system.

Kind of Study	Protein/Gene Studied	Population Studied	Main Findings	References
**Protein and gene expression studies**	GLT-1 and glutamate	Animal models (chronic unpredictable mild stress model of depression)	ECS down-regulated the expression of GLT-1 and elevated glutamate concentration in hippocampus.	[86]
	Glutamate receptor	Animal models	ECS reduced the expression of genes related to NMDA receptor, including glutamate receptor, in the frontal cortex.	[121]
	Glutamate receptor	Animal models	ECS changed GluR1 and NMDAR1 glutamate receptor subunit in the hippocampus.	[26]
	NMDA and glutamate	Animal models (depression models)	ECS ameliorated depressive symptoms and reversed the elevated hippocampal glutamate content and the down-regulation of an NMDA receptor.	[123]
	Glutamate	Animal models (depression models)	ECS increased hippocampal glutamate levels and the hyperphosphorylation of tau protein, related to glutamate metabolism.	[124]
	Glutamate	Clinical samples	ECT increased N-acetylaspartate and glutamine/glutamate levels in patients with TRD.	[125]
**Genetic studies**	*GRIK4* gene	Clinical samples	Variants in the *GRIK4* gene associated with ECT response in patients with TRD.	[129]
**Epigenetic studies**	NMDA signaling	Animal models	ECS increased HDAC2 brain expression and reduced the expression of genes related to NMDA.I It also decreased histone acetylation levels of H3 and/or H4.	[121]

ECS: electroconvulsive seizure; ECT: electroconvulsive therapy; GLT-1: glial glutamate transporter; GluR1: metabotropic glutamate receptor; *GRIK4*: glutamate receptor ionotropic kainate 4; HDAC2: histone deacetylase; NMDA: N-methyl-D-aspartate; SNP: single nucleotide polymorphism; TRD: treatment-resistant depression.

**Table 5 brainsci-11-01120-t005:** Main findings of ECS/ECT biological mechanisms of action related to the endocrine system.

Kind of Study	Protein/Gene Studied	Population Studied	Main Findings	References
**Protein studies**	Endocrine mediators	Animal samples	ECS increased ACTH levels in rodents.	[134]
	Endocrine mediators	Clinical samples	ECT exerted enhancing short-term effects on the release of hormones involved in the HPA axis, including ACTH, prolactin, and vasopressin.	[130,131,132,133]

ACTH: adrenocorticotropin; ECS: electroconvulsive seizure; ECT: electroconvulsive therapy; HPA: hypothalamic–pituitary–adrenal.

**Table 6 brainsci-11-01120-t006:** Main findings of ECS/ECT biological mechanisms of action related to the oxidative stress system and mitochondrial bioenergetics.

Kind of Study	Protein/Gene Studied	Population Studied	Main Findings	References
Protein studies	Oxidative stress system mediators	Animal samples	Decrease in oxidative damage parameters in the rat hippocampus.	[137]
	Oxidative stress system mediators	Animal samples	Decrease in lipid peroxidation and protein carbonyls in the rat hippocampus, cerebellum, and striatum after ECS.	[138]
	Oxidative stress system mediators	Animal samples	Increases in rat hippocampal and cerebellar SOD and GPX activities following ECS.	[139]
	Oxidative stress system mediators	Animal samples	Decrease in SOD and GPX activity in rats’ brains after ECS.	[140]
	Oxidative stress system mediators	Animal samples	Delayed oxidative damage after ECS in the rat hippocampus and striatum.	[141]
	Oxidative stress system mediators	Animal samples	Decrease in mitochondrial respiration and an increase in RNA oxidation in the rat brain tissue after ECS.	[142]
	Oxidative stress system mediators	Clinical samples	Reduction in serum total oxidant status values and an increase in total antioxidant status in patients with MDD.	[143]
	Oxidative stress system mediators	Clinical samples	SOD activity decreased after ECT in patients with bipolar disorders and MDD.	[144]

ECS: electroconvulsive seizure; ECT: electroconvulsive therapy; GPX: glutathione peroxidase; MDD: major depressive disorder; SOD: superoxide dismutase.

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
