# Peer review of "Molecular Biomarkers of Electroconvulsive Therapy Effects and Clinical Response: Understanding the Present to Shape the Future"

_brainsci, 2021, doi:10.3390/brainsci11091120_

Round 1

Reviewer 1 Report

The authors collected, categorized and summarized information available from the literature about different biological levels that are under investigation for the identification of molecular biomarkers of electroconvulsive therapy (ECT) effects. These observations were made mainly in treatment-resistant clinical cohorts with a diagnosis of an affective disorder. The concept and the rationale of the given work are very welcome, and the presented content and its discussion are very sound. Especially the title is well constructed and sounds catchy.

I found some minor issues that should be commented on and/or revised by the authors.

In the following, these minor issues are listed in a point-by-point manner:

Abstract:

  • In the abstract, cortisol is listed as one of the examples of “inflammatory markers“. This functional classification might be new to - at least - parts of the readership as cortisol is most often considered a major endocrine stress signalling molecule and an anti-inflammatory agent. I suggest using: ...ECT produces a reduction in inflammation-associated biomarkers such as".
  • From an ethical perspective, I recommend better use of "patients with MDD" instead of "MDD patients" (comparable to "HIV patients" vs. "patients with HIV").

  1. Aims of the review:
  • Besides the biological levels of interest discussed in the document, I find some suggested links also related to changes in bioenergetic states and biochemical energy supply and their possible changes following ECT treatment. These aspects are already represented in the reference list, but not included in the manuscript contentwise. Being a recently identified aspect in psychobiological and clinical biomarker research and a promising biomarker candidate,  I was wondering why the authors do not include mitochondrial bioenergetics and its changes in stress-related psychopathology also in the manuscript (at least in section 9).
  • The biomarker approaches listed in the manuscript are - besides the given genome-wide association studies and the gene expression profiling - following targeted (hypothesis-driven) analyses. It would be favourable to - at least- provide a short outlook to the advances of other untargeted approaches, for example from the fields of metabolomics, lipidomics or proteomics. I am aware of the fact that the available literature covering these methods in the field of ECT treatment is very limited, but communicating this limited availability might stimulate or even initiate more work. However, -Omics research on biomarkers of MDD is much more available and would be a valuable addition.  
  • As a general comment, it is important to clearly state in which samples, cells or tissues data was obtained from. This is of central importance in the case of research related to epigenetics. Please indicate in the text the biological origin of the listed observations by adding the information what kind of biomaterial was used in the experiments.
  • Page 3, line 128: The authors state that ECS was reported to enhance neurogenesis in the hippocampus of rats. In my opinion, it would be necessary to also state and relativize that the physiological function of the hippocampus alone does not allow to resolve the complexity of clinical symptoms of TRD on a functional or biomolecular scale. 
  • Additionally to the role of mitochondria in the bioenergetic supply of cells and tissues as well as the mediating role of inflammatory signalling, literature also provided evidence for an association between BDNF and other neuroplasticity markers and mitochondrial functioning in psychopathologies (e.g. 10.1074/jbc.P113.526129, 10.1093/ijnp/pyy022, 10.3389/fpsyt.2020.514658). I recommend adding this information to either a subsection of 4.1 or - at least - to section 9 ("Conclusion and future direction"). 
  • 4.3. Epigenetic studies: As mentioned previously, the robustness of observations made by in vitro-studies investigating biological changes on the level of epigenetics depends on the cell types or tissues used in the experiments and therefore biological changes that might affect the nature and composition of a biological sample (e.g. changes in immune cell-composition of whole blood) also need to be considered for the interpretation of results. Therefore, I ask the authors - especially for lines 223-243 - to add the information which cells were used in the given studies.
  • Page 6, line 282: The authors state that "a decrease in microglial cell number was found after ECS administration". It would be helpful to better understand the temporal character of this observation by adding the time frame between treatment and cell counting. This might be of interest especially to clinicians working with ECT.
  • Page 6, line 304: The information is given that "the kynurenine pathway plays a critical role in generating cellular energy". Again, I see a strong rationale to add the concept of changes in mitochondrial bioenergetics to this context. 
  • Page 7, line 309: The author list a study that was "conducted on patients with both unipolar and bipolar depression". I suggest changing "bipolar depression" to "bipolar disorders" to unify the nomenclature of the disease throughout the whole manuscript. 
  1. Conclusions and future directions: 
  • Most of the presented biomarkers are related to hypothesis-driven and therefore targeted biomarker discovery approaches. What about untargeted approaches, e.g. metabolomic, lipidomic or proteomic fingerprinting? The information about these available discovery approaches would strengthen the section on future directions as multivariate statistics combined with exploratory approaches and big data are essential to uncover biomolecular pathways and new candidates for clinically applicable biomarkers.
  • Emphasizing this issue for the last time, oxidative phosphorylation (OXPHOS) is the driving force of mitochondrial energy production and resulting adenosine triphosphate (ATP) is essential for the physiological functioning of all body cells. I recommend adding this aspect and its implications for stress-related psychopathologies, at least shortly, to the future directions section.

Author Response

Reviewer 1:

The authors collected, categorized and summarized information available from the literature about different biological levels that are under investigation for the identification of molecular biomarkers of electroconvulsive therapy (ECT) effects. These observations were made mainly in treatment-resistant clinical cohorts with a diagnosis of an affective disorder. The concept and the rationale of the given work are very welcome, and the presented content and its discussion are very sound. Especially the title is well constructed and sounds catchy.

I found some minor issues that should be commented on and/or revised by the authors.

In the following, these minor issues are listed in a point-by-point manner:

Abstract:

  • In the abstract, cortisol is listed as one of the examples of “inflammatory markers“. This functional classification might be new to - at least - parts of the readership as cortisol is most often considered a major endocrine stress signalling molecule and an anti-inflammatory agent. I suggest using: ...ECT produces a reduction in inflammation-associated biomarkers such as".

Answer: The sentence was modified for better clarification, as suggested (lines 27 and 28).

  • From an ethical perspective, I recommend better use of "patients with MDD" instead of "MDD patients" (comparable to "HIV patients" vs. "patients with HIV").

Answer: The writings were corrected both in the abstract and throughout the text and tables, as suggested.

  1. Aims of the review:
  • Besides the biological levels of interest discussed in the document, I find some suggested links also related to changes in bioenergetic states and biochemical energy supply and their possible changes following ECT treatment. These aspects are already represented in the reference list, but not included in the manuscript contentwise. Being a recently identified aspect in psychobiological and clinical biomarker research and a promising biomarker candidate, I was wondering why the authors do not include mitochondrial bioenergetics and its changes in stress-related psychopathology also in the manuscript (at least in section 9).

Answer: We thank the reviewer for the great suggestion. We have provided a dedicated section (section 9), entitled “Oxidative stress system and mitochondrial bioenergetics” and a relative table, including some selected articles, taking into consideration the research available in this theme.

  • The biomarker approaches listed in the manuscript are - besides the given genome-wide association studies and the gene expression profiling - following targeted (hypothesis-driven) analyses. It would be favourable to - at least- provide a short outlook to the advances of other untargeted approaches, for example from the fields of metabolomics, lipidomics or proteomics. I am aware of the fact that the available literature covering these methods in the field of ECT treatment is very limited, but communicating this limited availability might stimulate or even initiate more work. However, -Omics research on biomarkers of MDD is much more available and would be a valuable addition.  

Answer:

We have added this sparse availability of broader approaches on section 10, as perspectives for future studies.

  • As a general comment, it is important to clearly state in which samples, cells or tissues data was obtained from. This is of central importance in the case of research related to epigenetics. Please indicate in the text the biological origin of the listed observations by adding the information what kind of biomaterial was used in the experiments.

Answer: Biological materials used for the experiments were added in the sections, accordingly.

  • Page 3, line 128: The authors state that ECS was reported to enhance neurogenesis in the hippocampus of rats. In my opinion, it would be necessary to also state and relativize that the physiological function of the hippocampus alone does not allow to resolve the complexity of clinical symptoms of TRD on a functional or biomolecular scale. 

Answer: This valuable observation was added in the text, on page 4, lines 136-140.

  • Additionally to the role of mitochondria in the bioenergetic supply of cells and tissues as well as the mediating role of inflammatory signalling, literature also provided evidence for an association between BDNF and other neuroplasticity markers and mitochondrial functioning in psychopathologies (e.g. 10.1074/jbc.P113.526129, 10.1093/ijnp/pyy022, 10.3389/fpsyt.2020.514658). I recommend adding this information to either a subsection of 4.1 or - at least - to section 9 ("Conclusion and future direction"). 

Answer: The relationship between BDNF and oxidative stress parameters was briefly added in the Neurotrophic system section (page 4, lines 152-155) and a general discussion about oxidative stress metabolism was added in a separate section (Section 9: Oxidative stress system and mitochondrial bioenergetics).

  • 4.3. Epigenetic studies: As mentioned previously, the robustness of observations made by in vitro-studies investigating biological changes on the level of epigenetics depends on the cell types or tissues used in the experiments and therefore biological changes that might affect the nature and composition of a biological sample (e.g. changes in immune cell-composition of whole blood) also need to be considered for the interpretation of results. Therefore, I ask the authors - especially for lines 223-243 - to add the information which cells were used in the given studies.

Answer: Biological materials used for the experiments were added in the sections, accordingly.

  • Page 6, line 282: The authors state that "a decrease in microglial cell number was found after ECS administration". It would be helpful to better understand the temporal character of this observation by adding the time frame between treatment and cell counting. This might be of interest especially to clinicians working with ECT.

Answer: The temporal information was added, accordingly, with an observation of possible clinical translation of the temporal data observed in animals (page 7, lines 295-301).

  • Page 6, line 304: The information is given that "the kynurenine pathway plays a critical role in generating cellular energy". Again, I see a strong rationale to add the concept of changes in mitochondrial bioenergetics to this context. 

Answer: A discussion about oxidative stress metabolism was added in a separate section (Section 9: Oxidative stress system and mitochondrial bioenergetics).

  • Page 7, line 309: The author list a study that was "conducted on patients with both unipolar and bipolar depression". I suggest changing "bipolar depression" to "bipolar disorders" to unify the nomenclature of the disease throughout the whole manuscript. 

Answer:  Modifications were made in order to clarify the nomenclature used throughout the manuscript.

  1. Conclusions and future directions: 
  • Most of the presented biomarkers are related to hypothesis-driven and therefore targeted biomarker discovery approaches. What about untargeted approaches, e.g. metabolomic, lipidomic or proteomic fingerprinting? The information about these available discovery approaches would strengthen the section on future directions as multivariate statistics combined with exploratory approaches and big data are essential to uncover biomolecular pathways and new candidates for clinically applicable biomarkers.

Answer: In section 10 we have added the importance of broader approaches and their sparse availability, with the necessity of further studies, as perspectives for future study designs.

  • Emphasizing this issue for the last time, oxidative phosphorylation (OXPHOS) is the driving force of mitochondrial energy production and resulting adenosine triphosphate (ATP) is essential for the physiological functioning of all body cells. I recommend adding this aspect and its implications for stress-related psychopathologies, at least shortly, to the future directions section.

Answer: A discussion about oxidative stress metabolism was added in a separate section (Section 9: Oxidative stress system and mitochondrial bioenergetics).

Reviewer 2 Report

The authors present a well-written narrative review aiming to summarize current molecular studies regarding ECT. This is important work, because – as the authors mention in their discussion – “ECT is the most effective treatment for severe and refractory depression but, at the same time, one of the most controversial and misunderstood treatments,” [and] “increasing the knowledge about its mechanisms could help reduce its stigmatization and false beliefs about it.” The authors report extensively several interesting molecular mechanisms, which might support others (and the authors) to formulate important hypotheses for further studies regarding human psychopathology and working mechanisms of ECT.

I have two comments:

1) No description was included of the specific results of the systematic literature search (e.g., total established studies in the search, number of human and animal studies, number of exclusions, reasons for exclusion, etc.), which makes it difficult for the readers to follow the selection process. Maybe, including a flow chart would be helpful to provide insight in this process;

2) Some more discussion could follow from the gathered literature. For example, some questions raised when I read the manuscript, which might be discussed in little more detail:

Which molecular mechanisms seemed fruitful for further study and why (or why not)?

Which overlaps could be made with other domains of research (e.g., epilepsia studies, neuropsychology, neuroimaging), because more integration with such other domains might be inspiring to the readers?

Which other (clinical) phenomena might be interesting in respect of the found molecular mechanisms (e.g., neurophysiological measures, brain perfusion measures, seizure threshold, seizure duration), because - maybe - some molecular mechanisms might influence these phenomena, distinct of their influence on clinical effectiveness of ECT.

Author Response

Reviewer 2:

The authors present a well-written narrative review aiming to summarize current molecular studies regarding ECT. This is important work, because – as the authors mention in their discussion – “ECT is the most effective treatment for severe and refractory depression but, at the same time, one of the most controversial and misunderstood treatments,” [and] “increasing the knowledge about its mechanisms could help reduce its stigmatization and false beliefs about it.” The authors report extensively several interesting molecular mechanisms, which might support others (and the authors) to formulate important hypotheses for further studies regarding human psychopathology and working mechanisms of ECT.

I have two comments:

1) No description was included of the specific results of the systematic literature search (e.g., total established studies in the search, number of human and animal studies, number of exclusions, reasons for exclusion, etc.), which makes it difficult for the readers to follow the selection process. Maybe, including a flow chart would be helpful to provide insight in this process;

Answer: The present review is not a systematic literature review but a narrative one. We have added further information in the methods section (page 3, lines 102 to 105). 

2) Some more discussion could follow from the gathered literature. For example, some questions raised when I read the manuscript, which might be discussed in little more detail:

Which molecular mechanisms seemed fruitful for further study and why (or why not)?

Answer: To further address this important question, perspectives about future directions for study designs were added in section 10, mainly regarding information that could be gathered from future multi omics approaches, instead of individual or hypothesis driven molecular mechanisms. 

Which overlaps could be made with other domains of research (e.g., epilepsia studies, neuropsychology, neuroimaging), because more integration with such other domains might be inspiring to the readers?

Answer: General perspectives of integrated studies (with molecular, structural and functional approaches to better understand the biological mechanisms of ECT) applied to other neuropsychiatric disorders, including epilepsy and schizophrenia, for example, were added in the section 10.

Which other (clinical) phenomena might be interesting in respect of the found molecular mechanisms (e.g., neurophysiological measures, brain perfusion measures, seizure threshold, seizure duration), because - maybe - some molecular mechanisms might influence these phenomena, distinct of their influence on clinical effectiveness of ECT.

Answer: Thanks to the reviewer for this relevant clinical comment. Unfortunately, the literature about this issue is scarce and inconsistent. We added a sentence as a further limitation concerning this point in section 10.